# Thangka Image Captioning Based on Semantic Concept Prompt and Multimodal Feature Optimization

**DOI:** 10.3390/jimaging9080162

**Published:** 2023-08-16

**Authors:** Wenjin Hu, Lang Qiao, Wendong Kang, Xinyue Shi

**Affiliations:** 1School of Mathematics and Computer Science, Northwest Minzu Univsersity, Lanzhou 730030, China; 2Key Laboratory of China’s Ethnic Languages and Information Technology of Ministry of Education, Northwest Minzu University, Lanzhou 730030, China; qiaolang1997@gmail.com (L.Q.); gg15163771981@gmail.com (W.K.); shixinyue_jiayou@163.com (X.S.)

**Keywords:** image captioning, Thangka, deep learning, visual concepts, knowledge distillation

## Abstract

Thangka images exhibit a high level of diversity and richness, and the existing deep learning-based image captioning methods generate poor accuracy and richness of Chinese captions for Thangka images. To address this issue, this paper proposes a Semantic Concept Prompt and Multimodal Feature Optimization network (SCAMF-Net). The Semantic Concept Prompt (SCP) module is introduced in the text encoding stage to obtain more semantic information about the Thangka by introducing contextual prompts, thus enhancing the richness of the description content. The Multimodal Feature Optimization (MFO) module is proposed to optimize the correlation between Thangka images and text. This module enhances the correlation between the image features and text features of the Thangka through the Captioner and Filter to more accurately describe the visual concept features of the Thangka. The experimental results demonstrate that our proposed method outperforms baseline models on the Thangka dataset in terms of BLEU-4, METEOR, ROUGE, CIDEr, and SPICE by 8.7%, 7.9%, 8.2%, 76.6%, and 5.7%, respectively. Furthermore, this method also exhibits superior performance compared to the state-of-the-art methods on the public MSCOCO dataset.

## 1. Introduction

Thangka is an invaluable intangible cultural heritage in the world’s folk art, which refers to the “painting that records Buddhist teachings” in Tibetan [1,2]. Thangka paintings feature rich subject matter and complex techniques, depicting Buddhist doctrines and philosophical thoughts through meticulous drawing and vibrant colors, making them an indispensable part of Tibetan culture. However, accurately describing their characteristics and connotations requires individuals with rich artistic and domain knowledge, which can be time-consuming and challenging. Image captioning models based on deep learning can enable computers to automatically learn and describe the content of Thangka images, which can facilitate our understanding of Thangka images, promote the dissemination of excellent traditional Chinese culture, and have a positive impact on supplementing and expanding research on image captioning.

Image captioning is a highly challenging task that requires transforming the complex visual information of an image into a natural language description while considering the sentence’s grammatical structure and contextual information. The dataset’s quality, the model’s complexity, the task’s difficulty, and the choice of evaluation metrics are all related to the performance of the image caption model. Despite the significant progress made by deep learning-based image captioning methods [3,4,5], there is currently no feasible solution for generating Chinese captions for Thangka images, mainly due to the following three reasons. First, Thangka images contain rich content that needs to be described, including various information such as Buddha types, morphology, and textual symbols, which existing models have difficulty fully learning, leading to incomplete generated captions. Second, Thangka images have complex backgrounds, including details such as patterns, symbols, and designs, and the accuracy of existing image captioning methods in generating captions for these details is low. Third, the small size of the Thangka dataset makes it difficult to adequately train existing image captioning methods, which affects the quality of the generated captions.

Recently, image captioning models based on contrastive language images [6,7,8] have been widely used in multimodal tasks of images and texts with remarkable results. These models train a language–image embedding space through contrastive learning, where matching image–text embeddings are brought closer together. In contrast, non-matching ones are pushed further apart, improving the correlation between images and text. Therefore, this method also has great potential in generating Chinese captions for Thangka images. However, it still faces two challenges in producing satisfactory Chinese captions for Thangka images:

**Challenge 1:** Compared to other image captioning tasks, Thangka image captioning involves rich and abstract information, such as cultural connotations and domain-specific knowledge. However, existing algorithms have limitations in learning visual concepts and can only grasp specific visual features of Thangka images, resulting in a low richness of generated Thangka captions.

**Challenge 2:** Although existing contrastive language–image methods can effectively align the features of Thangka images and text, they cannot verify whether the image–text pairs in the training dataset are correctly matched. Noisy data during training can significantly reduce the model’s accuracy.

To address the challenges in Thangka image captioning, this paper proposes a Semantic Concept Prompt and Multimodal Feature Optimization Network (SCAMF-Net), as shown in Figure 1. The proposed network aims to generate more accurate and richer captions for Thangka images, encompassing not only specific visual feature information, such as character recognition and behavior description of Thangka figures, but also various other aspects of information, such as Thangka cultural connotations and background knowledge. The proposed method in this paper consists of two parts:

**First (solution for Challenge 1)**, we proposed a Semantic Concept Prompt (SCP) module consisting of a learnable vector and traditional text features. We use the learnable vector to model the context words of the prompt, which can effectively utilize multi-aspect information in the Chinese captions of Thangka images and integrate more information sources into the prompt, providing rich contextual information for the image captioning model.

**Second (solution for Challenge 2)**, we proposed a Multimodal Feature Optimization (MFO) module based on Transformer architecture. The module can accurately distinguish whether the image–text pairs match by introducing cross-attention layers and an Image–Text Matching (ITM) loss to construct the Filter. The Captioner utilizes causal attention layers and introduces a Language Modeling (LM) loss to enrich the text features of Thangka further and optimize the data source, improving the relevance between Thangka images and the generated Chinese captions. The MFO module transfers the knowledge of the complex image captioning model to a simpler model, avoiding overfitting the training data and improving the model’s understanding of Thangka and computational efficiency.

The experimental results demonstrate that our proposed SCAMF-Net provides a highly effective image captioning method for generating accurate Thangka captions. The contributions of this study are as follows:1.We proposed the Semantic Concept Prompt (SCP) method, which introduces context prompts on top of traditional text features. This enables the model to learn abstract concepts embedded in Thangka images, such as cultural connotations and domain-specific knowledge, within the closed visual concept space and enriches the generated captions. The SCP method improves the accuracy and richness of generated Thangka captions.2.We designed the Multimodal Feature Optimization (MFO) module, which includes the Filter that removes the noisy data in the Thangka image–text pairs during the model training process and the Captioner that enriches the text features of Thangka images and optimizes the data source, thereby improving the accuracy and stability of the image captioning model and making the generated Thangka image captions better match the Thangka images.3.This paper proposed a Thangka image captioning method based on Semantic Concept Prompt and Multimodal Feature Optimization, called SCAMF-Net. We also construct a new Thangka dataset, providing essential resources for further research.

## 2. Related Work

### 2.1. Visual Language Pre-Training

Deep learning-based image captioning methods improve the correlation between images and texts and enhance model performance by optimizing image and text feature extractors. However, despite the increasing maturity of methods for optimizing feature extractors [9,10,11,12], training models on small datasets remains challenging and may not provide sufficient training for the model. Visual Language Pre-Training (VLP) involves pre-training deep learning models on large-scale datasets to transfer learned knowledge to downstream tasks, thereby improving model efficiency and performance. As a result, more researchers are adopting Visual Language Pre-training to enhance model performance. VLP can be categorized into two types: (1) language modeling and (2) image–text contrastive learning.

#### 2.1.1. Language Modeling Methods

Anderson et al. [13] improved the extraction of image feature information and enhanced the correlation between image and text features by incorporating object detection algorithms. However, their model relies on pre-training to extract region features, such as Faster-RCNN, which reduces the model’s scalability as it requires additional target annotation data. Dosovitskiy et al. [14] proposed the Visual Transformer (ViT) model, which divides an image into small blocks for processing, reducing the computational complexity and time. Liu et al. [15] introduced the Swin Transformer, which extracts visual features through windows and sliding windows, interacts with adjacent windows by moving the window, and globally models feature information by connecting the upper and lower cross windows. However, language modeling-based methods require more profound model training, are computationally intensive, and are less flexible. Khare et al. [16] proposed Multimodal Bert (MMBert), which is a Bert-based multimodal learning model capable of effectively handling various modalities of data, including text, images, and videos. Lu et al. [17] introduced Vision-and-Language Bert (ViLBert), which uses two independent Bert encoders to process image and text information and fuses them using attention mechanisms. Tan et al. [18] proposed Language-and-Visual Matching with Hierarchical Transformers (LXMERT), which combines VisualBert and Bert to achieve joint encoding of visual and language information. Chen et al. [19] proposed Universal Image–Text Representation (UNITER), which utilizes a novel cross-modal alignment strategy that better handles the relationship between images and text. Specifically, the model introduces a multi-task learning strategy that facilitates alignment and interaction between images and text by simultaneously learning multiple tasks.

#### 2.1.2. Image–Text Contrastive Learning Methods

In recent years, image–text contrastive learning-based methods have demonstrated state-of-the-art performance in image captioning. The core idea is to reduce the distance between matched image–text embeddings and expand the distance between unmatched embeddings, creating an embedding space aligned with the image and text. Under the guidance of unsupervised contrastive loss, the model can automatically learn semantic relevance. Zhou et al. [20] used a multi-layer Transformer network for encoding and decoding and pre-trained it on many image–text pairs. However, this method only uses image region features and text features as inputs to the model for pre-training and learns image–text semantic alignment simply and crudely through self-attention. Radford et al. [21] proposed the Contrastive Language–Image Pre-training (CLIP) model, which was trained on over 400 million image–text pairs, generating a rich shared semantic latent space for visual and textual data. Mokady et al. [22] used a simple mapping network and fine-tuned language model to generate image captions by treating CLIP encoding as a prefix. Zeng et al. [23] proposed the Socrates Model (SM), which aggregated multiple mainstream image–text contrastive models into a large model to find a unified embedding space for multiple modalities to complete the task. Although these methods have greatly improved the model’s performance, data due to incorrect matching of image–text pairs during model training can reduce the model’s performance.

### 2.2. Prompt Learning

Prompt learning was initially applied in natural language processing (NLP), bringing downstream tasks closer to pre-trained models using prompts. It can perform tasks with zero-shot or few-shot learning by inputting prompts as context to the pre-trained model. Ref. [24] demonstrated that adjusting only soft prompt embeddings at each layer is sufficient to achieve good performance of pre-trained models in natural language generation. Ref. [25] uses text mining to generate a set of candidate prompts to select the best prompt with the highest training accuracy.

In computer vision, prompt learning is a new research direction that has only recently been explored. The prompt learning method for visual language pre-training models is mainly used for models similar to the CLIP model, where prompts for text features are used to optimize model performance. Ref. [26] demonstrated that when using large-scale visual language pre-training, the visual encoder used for prompt learning is sufficient for multimodal zero-shot learning. To further advance this approach, ref. [27] proposed Flamingo, which achieves context-aware learning and state-of-the-art performance in cross-modal tasks in scenes under zero-shot conditions. Ref. [28] proposed a method that uses learnable prompts to generate image caption content. They built a key-value object memory using similar caption objects and an external detection model to address the issue of generating novel descriptions without new object descriptions during training. In contrast, this paper focuses on leveraging existing caption content and further utilizing the details and abstract concepts within it to generate more accurate and detailed image captions.

Despite the benefits of prompt learning in making pre-trained models and downstream tasks more relevant, setting prompts requires specialized knowledge and takes a significant amount of time to adjust. Even minor changes to prompts can have a significant impact on performance.

### 2.3. Knowledge Distillation

Knowledge distillation is primarily a model compression method that uses a teacher–student model structure. In this model, the teacher model is usually a pre-trained model with a relatively complex structure, and the student model acquires knowledge from the teacher model through distillation training. Most methods [29,30,31] directly use pre-trained models as teacher models and improve the performance of the student model at the cost of slight performance loss, thereby enhancing the model’s generalization ability. In recent years, visual language pre-training-based methods have been increasingly used in downstream tasks. Refs. [32,33] used this type of contrastive language model as a teacher model and reduced the model’s parameter size through distillation, thereby improving performance on visual language tasks.

Although the above methods have improved the model’s generalization ability through knowledge distillation, generating inaccurate and unclear captions is still a challenge due to the significant cross-modal gap between images and text.

## 3. Methodology

To address the issues of low accuracy and insufficient content richness in generating Chinese captions for Thangka images using traditional deep learning-based algorithms, we propose the SCAMF-Net image caption network based on the contrastive language–image method, as illustrated in Figure 1. Section 3.1 presents the SCP module’s specific structure and the method for improving the Thangka text feature extraction ability. Section 3.2 describes the MFO module, which includes the Captioner and Filter structures, and the method for optimizing the contrastive language–image embedding space. Finally, Section 3.3 introduces the loss function used by the model.

### 3.1. Semantic Concept Prompt (SCP)

Prompt learning has become a popular approach in natural language processing for handling multimodal tasks involving visual language, producing remarkable results. Large-scale visual language pre-training models such as CLIP allow for zero-shot transfer to downstream tasks if prompts are added after text feature extraction. However, setting prompts manually for image caption tasks is labor-intensive and time-consuming, especially for the Chinese caption of Thangka images. Additionally, standard prompt learning methods strongly depend on data and struggle with managing complex scenes in image caption processing. To address these challenges, we propose the SCP method, which employs end-to-end learning to model continuous vectors of context words from the data. SCP eliminates the requirement for manual tuning, leading to more precise and more affluent content generation by the Thangka image caption model.

Figure 1b illustrates the SCP module, which begins by extracting the text features of the Thangka caption using a text encoder. The extracted features are then concatenated with the Learnable Context to form the [CLS] text feature, which is input into the Transformer. The multi-head attention mechanism is applied to extract features and model relationships, thereby improving the correlation between image features and text features and capturing the text information optimally. In the Thangka image caption task, the SCP model better learns Thangka’s rich domain knowledge and cultural connotations, resulting in a more diverse generated caption. The SCP method utilizes a set of learnable vectors called Learnable Context to model the context of prompts and minimizes the classification loss during model training to optimize the process. The prompt vector s is designed as follows: (1)$$s={\left[V\right]}_{1}{\left[V\right]}_{2}\ldots {\left[V\right]}_{M}$$
where each ${\left[V\right]}_{m}\left(m\in \left\{1,\ldots ,M\right\}\right)$ is a vector with the same dimension as the word embedding, and *M* is a hyperparameter that represents the context length, with a default size of 8. The prompt token *s* is concatenated with the text embedding *t* obtained from traditional methods to form a new text feature [CLS]. The model obtains a new classification weight vector that represents visual concepts. The prediction probability is then calculated.
(2)$$p\left(y=i|x\right)=\frac{\exp\left(<g\left(s_{i}\right),f>/\tau \right)}{\sum_{j=1}^{K}\exp\left(<g\left(s_{j}\right),f>/\tau \right)}$$
where the class token in each prompt $t_{i}$ is replaced by the corresponding *i*-th word embedding.

Summary of advantages: The SCP method offers several advantages over traditional prompt learning methods. Firstly, it eliminates the laborious and time-consuming process of manually setting prompts, making the Thangka image caption task more efficient and scalable. Additionally, the SCP method introduces additional domain knowledge by opening up visual concepts when processing Chinese captions for Thangka images. This approach unleashes the model’s potential, avoiding obvious grammatical errors and capturing various types of information in the Thangka images, including rich background knowledge and cultural connotations. As a result, the generated captions are more diverse and informative, enabling a better understanding of Thangka images.

### 3.2. Multimodal Feature Optimization (MFO)

Despite significant progress in image captioning through visual language pre-training methods, the vast cross-modal differences between images and text still pose a challenge, resulting in less accurate captions for Thangka image Chinese caption tasks. Additionally, most training datasets are manually curated and may contain subjective judgments, while the text may include content unrelated to the images, leading to suboptimal data sources. To enhance the efficiency and generalization ability of the image caption model, we employed knowledge distillation methods [34]. However, standard knowledge distillation methods may need more information loss and more diversity in generated captions. Therefore, we propose the MFO module, which dynamically extracts knowledge from the Thangka image caption model and transfers it to the student model to improve the accuracy and diversity of the generated captions while maintaining model performance.

As shown in Figure 1c, given a Thangka image–text pair, we use the Vision Transformer [14] as the image encoder to extract the image features of the Thangka and the BERT [35] as the text encoder to extract the text features of the Thangka captions. Both are based on the Transformer, and the core of the Transformer is the self-attention mechanism, which is used to reduce complexity and computational cost.
(3)$$\mathrm{Attention}\left(Q,K,V\right)=\mathrm{softmax}\left(\frac{QK^{T}}{\sqrt{d_{k}}}\right)V$$

The input to the attention mechanism includes *Q* (query), *K* (key), and *V* (values), and the output is a weighted average vector that represents the importance of different positions in the input sequence for the model output. Specifically, for each position in the input sequence, the attention model calculates a weight that represents the contribution of that position to generating the current output. Then, these weights are multiplied by the corresponding position vectors and summed to obtain a weighted average vector, which is the output of the attention mechanism. By reducing the distance between the matched image–text embeddings and increasing the distance between mismatched embeddings, we ultimately form an aligned embedding space for image–text pairs. We use a similarity function to achieve this goal, where f and g are functions that map the [CLS] embeddings obtained from the SCP module to a normalized low-dimensional space. By learning the similarity function, we calculate the similarity score between image–text pairs, which allows similar pairs to have higher scores. Inspired by MoCo [36], we use two queues to store the closest image–text representations in the encoder, and the normalized features of the encoder are denoted as $g_{v}^{\prime }\left(v_{\mathrm{cls}}^{\prime }\right)$ and $g_{w}^{\prime }\left(w_{\mathrm{cls}}^{\prime }\right)$. For each image and text, we calculate the softmax-normalized similarity scores between the image and text and between the text and image as follows:(4)$$p_{m}^{i2t}\left(I\right)=\frac{\exp\left(s\left(I,T_{m}\right)/\tau \right)}{\sum_{m=1}^{M}\exp\left(s\left(I,T_{m}\right)/\tau \right)},\;p_{m}^{t2i}\left(T\right)=\frac{\exp\left(s\left(T,I_{m}\right)/\tau \right)}{\sum_{m=1}^{M}\exp\left(s\left(T,I_{m}\right)/\tau \right)}$$
where $s\left(I,T\right)=g_{v}{\left(v_{\mathrm{cls}}\right)}^{⊤}g_{w}^{\prime }\left(w_{\mathrm{cls}}^{\prime }\right)$ and $s\left(T,I\right)=g_{w}{\left(w_{\mathrm{cls}}\right)}^{⊤}g_{v}^{\prime }\left(v_{\mathrm{cls}}^{\prime }\right)$, and τ is a learnable parameter.

Next, the text feature vector [Encode] is fed into the Filter structure, where the image feature vector is introduced through a cross-attention layer to obtain the similarity scores for the image–text pairs. The matched image–text pairs are selected by applying a threshold to filter out the mismatched noise data. Then, the captioner is used as the decoder to enrich the text features of the Thangka dataset by using the [Decode] token as the first input. Specifically, at each time step, we use the vector representation of the previously generated word as the input for the next time step and combine the features of the image encoding with the context of the text using the attention mechanism.

Summary of Advantages: In addition to enhancing the generalization ability of the image caption model, the MFO module further improves the correlation between Thangka image and text features through the filter and generator, making it more suitable for Chinese caption tasks for Thangka images. During model training, the MFO module optimizes the embedding space and stability of images and text to ensure a more accurate caption of Thangka image content, including features related to the type of Buddha, special symbols, and actions.

### 3.3. Loss Function

The choice of the loss function for the model is crucial in determining the direction of training and has a significant impact on the results of Chinese caption tasks for Thangka images. We aim to generate accurate and rich Thangka content to help people better understand the information conveyed in Thangka images. When selecting the loss function, it is essential to carefully consider the model’s performance and generalization ability and continuously explore ways to improve the model’s performance and generation effects. Therefore, we adopted three loss functions that are commonly used in the optimization of multimodal features: (1) image–text contrastive loss, which encourages matched image–text pairs to be close in the embedding space and pushes mismatched pairs away; (2) image–text matching loss, which measures the similarity between the generated text and the ground-truth captions; and (3) language modeling loss, which maximizes the probability of generating the correct next word in the sequence.

**Image–Text Contrastive Loss.** Image–text contrastive loss is a commonly used loss function for training multimodal image and text models. This loss function contrasts a set of positive and negative samples to learn the semantic relationship between images and text, thereby improving the model’s expressive power. We denote the manually labeled image and text similarity as $y^{i2t}\left(I\right)$ and $y^{t2i}\left(T\right)$ and the model-generated image and text similarity as $p^{i2t}\left(I\right)$ and $p^{t2i}\left(T\right)$, where the probability of an opposing pair is 0. The probability of a cheerful pair is 1. The image–text contrastive loss is defined as the cross-entropy H between *p* and *y*:(5)$$L_{\mathrm{itc}}=\frac{1}{2}E_{(I,T)∼D}\left[H\left(y^{i2t}\left(I\right),p^{i2t}\left(I\right)\right)+H\left(y^{t2i}\left(T\right),p^{t2i}\left(T\right)\right)\right]$$

**Image–Text Matching Loss.** The image–text matching loss uses the output embedding of the multimodal encoder as a joint representation of the image–text pair. It uses softmax and fully connected layers to compute the loss. The image–text matching loss is defined as: (6)$$L_{\mathrm{itm}}=E_{(I,T)∼D}H\left(y^{\mathrm{itm}},p^{\mathrm{itm}}\left(I,T\right)\right)$$
where $y^{\mathrm{itm}}$ represents a one-hot vector to determine whether the label is correct or not. For a batch of image–text pairs, which is sampled by contrastive similarity, texts more similar to images are more likely to be sampled.

**Language Modeling Loss.** The language modeling loss aims to generate a textual caption of a given image, which optimizes the cross-entropy loss and trains the model to maximize the likelihood of the text in an autoregressive manner, using the image and contextual text to predict mask words. We randomly mask the input tokens with a 15% probability and replace them with the mask. Assuming that *T* is the mask text and $p^{\mathrm{msk}}$ is the prediction probability of the mask, and the loss of language modeling is as follows:(7)$$L_{\mathrm{lm}}=E_{(I,\hat{T})∼D}H\left(y^{\mathrm{msk}},p^{\mathrm{msk}}\left(I,\hat{T}\right)\right)$$
where $y^{\mathrm{msk}}$ is the one-hot vector with a probability of 1 of the positive label.

## 4. Experimental Results and Analysis

### 4.1. Dataset and Experiment Setup

We used the Thangka dataset and the MSCOCO 2014 dataset to evaluate the effectiveness of the proposed method. Firstly, we constructed a Thangka image caption dataset by obtaining Thangka images from the internet, including various types such as Buddha, Bodhisattva, Vajra, and Deity Mother. Specifically, there are 631 images of Buddha, including Amitabha, Maitreya, Shakyamuni, and Longevity Buddha; 306 images of Bodhisattva and Deity Mother; 330 images of Vajra, including White Tara, Prajna Paramita, and Wealth Deity Mother; and 270 images of Bodhisattva, including Ksitigarbha and Maitreya. Each image is manually annotated with a corresponding Chinese caption, which includes the type and action of the main deity, high-level semantic concepts, and the symbolic meaning of special symbols, with a length of around 100 words. To enhance the data’s diversity and the model’s robustness, we applied data augmentation techniques such as random cropping, rotation, flipping, and scaling to the Thangka images, increasing the number of images to 3974. Finally, the dataset was split into a training set with 3242 images, a validation set with 396 images, and a test set with 336 images. To further validate the generalization ability of the proposed model, we also conducted experiments on the MSCOCO dataset, which includes 123,287 images with English captions of various scenes. We adopted the split method of Karpathy [37], which includes 113,287 training images, 5000 validation images, and 5000 test images, each with five English captions, to evaluate the generalization ability and effectiveness of the Thangka image Chinese caption model proposed in this paper.

The experiments were conducted on a 64-bit Ubuntu 18.04 system using the PyTorch deep learning framework to build and train the model. The hardware configuration includes an Intel(R) Xeon(R) Gold 6130 CPU @ 2.10 GHz and an Nvidia RTX 2080Ti graphics card with 11 GB of memory. In the preprocessing stage, the image size is 384 × 384, and a pre-trained image encoder is used for feature extraction. The word vector’s dimension is 768, and the maximum length of the generated caption text is 100. The batch size of the input data is set to eight, and dropout regularization is used to improve the model’s generalization ability, with a dropout value of 0.1. The training optimizer is set to Adam, with a weight decay of 0.05 and a learning rate of 3 $\times 10^{-4}$, with a linear decay rate of 0.85. A beam search with a beam value of three is used during training.

### 4.2. Evaluation Indicators

At present, there are several evaluation indexes used for image caption, including Bilingual Evaluation Understudy (BLEU) [38], Metric for Evaluation of Translation with Explicit Ordering (METEOR) [39], Recall-Oriented Understudy for Gisting Evaluation (ROUGE) [40], Consensus-based Image caption Evaluation (CIDEr) [41], and Semantic Propositional Image Caption Evaluation (SPICE) [42]. These evaluation indexes emphasize different aspects of sentence evaluation. BLEU and METEOR were initially proposed as evaluation indexes for machine translation tasks, ROUGE was proposed for automatic summarization tasks, and CIDEr and SPICE were developed for image caption evaluation.

In BLEU, the phrase n-gram in the candidate sentence is adapted to match the phrases of the same length in the reference sentence to measure the similarity of the two sentences. The n-gram unit group is used to weigh the accuracy of the word, where n is the number of words used for word matching. The higher the n-gram, the smoother the generated sentences.

METEOR was proposed because the recall was found to be of great significance in the performance of image captioning. In contrast to precision-focused evaluation criteria such as BLEU, METEOR considers the influence of roots, affixes, and synonyms on the caption results. As a result, its results are more relevant to human judgment.

ROUGE is an evaluation index based on a recall similarity measure used to assess the adequacy and fluency of machine translation. It is primarily used to evaluate machine translation and text summarization tasks.

CIDEr evaluates the consistency between a generated caption and reference captions by calculating the TF-IDF weight of n-tuples in the corpus. It computes weighted n-gram similarity scores using TF-IDF, which means that its scores can exceed one. The unit of CIDEr is a score representing the similarity between the generated image captions and the human-generated reference captions. Although the score range of CIDEr is between 0 and infinity, it typically falls between 0 and 6.

SPICE encodes objects, attributes, and relationships in the title, constructs a scene graph, and performs graph-level matching through the semantic representation of the graph. Compared to evaluation criteria based on n-gram or frequency, the graph structure in SPICE is more complex, and the relationship between words is more abundant. Experiments have shown that SPICE’s assessment of sentence similarity is closer to human-level judgment.

### 4.3. Quantitative Evaluation

We conducted experiments on the Thangka dataset and COCO dataset to evaluate the effectiveness of the proposed SCAMF-Net method and compared it with several existing mainstream image captioning models, including Bottom-Up and Top-Down Attention (BUTD) [13], Attention on Attention (AoA) [43], Object-Semantics Aligned Pre-training for Vision-Language Tasks (Oscar) [44], Bidirectional Long Short-Term Memory Network (Bi-LSTM) [45], and Object-Semantics Aligned Pre-training for Vision-Language Tasks (LEMON) [46], etc.

For the Thangka Chinese caption task, we tokenized the Thangka caption content using Chinese word embeddings. The experimental results in Table 1 indicate that our proposed SCAMF-Net outperforms the current mainstream models in various evaluation metrics on the Thangka dataset. In addition, VGG, Inception, and EfficientNet V2, which are CNN-based architectures, perform poorly in handling the task of Thangka image description. These methods primarily rely on convolutional layers for feature extraction, which poses difficulties in capturing long sequence dependencies and suffers from the issue of gradient vanishing. In contrast, Realformer and T5 utilize a Transformer-based architecture that effectively captures the contextual relationships between words or tokens in long sequences through a self-attention mechanism. As a result, they demonstrate excellent performance in the image caption.

The above analysis shows that the proposed SCAMF-Net method has significant advantages over other methods. Specifically, the SCP module in SCAMF-Net obtains more text feature space by concatenating prompt tokens and text features, enabling the image captioning model to generate more diverse Thangka content, improving the semantic similarity between generated text and manually annotated Thangka content, and significantly improving the CIDEr evaluation metric. Additionally, the MFO module distills and extracts Thangka knowledge learned by the baseline model through the Filter and Captioner. Specifically, the Filter filters out mismatched data in the Thangka image–text pairs, and the Captioner optimizes the text features in the Thangka dataset, improving the correlation between Thangka images and text, thereby increasing the frequency similarity of vocabulary between generated text and reference text and significantly improving the BLEU evaluation metric based on accuracy.

To further verify the stability and generality of the proposed method, we conducted experiments on the publicly available COCO dataset, and the results are shown in Table 2. Our method outperforms other advanced methods in the four evaluation metrics, namely BLEU-4, METEOR, CIDEr, and SPICE. Higher scores in these metrics indicate that the generated image captions are more similar to human captions in the dataset, making them easier to understand and accept. As a result, our method can produce more accurate, vivid, and natural image captions and has higher stability and generality, making it suitable for different types of image captioning tasks and achieving good experimental results.

### 4.4. Qualitative Evaluation

To qualitatively demonstrate the effectiveness of the SCAMF-Net model, we compared it with current advanced models and analyzed them in conjunction with manually annotated captions. As the Clipcap model performed well in quantitative experimental results, we mainly compared and analyzed the Clipcap and SCAMF-Net methods in the qualitative experiment. The results showed that compared with current advanced methods, the SCAMF-Net model performed better in extracting semantic correlations between Thangka images and texts and could more accurately describe the details of Thangka images and the relationships between targets within the images.

For example, the Thangka caption by Clipcap lacked detail and expression of cultural connotation information in Figure 2a. In contrast, the caption generated by the proposed SCAMF-Net method was more detailed and comprehensive, not only including the characteristics of the Thangka image itself but also reflecting the knowledge contained in the image more thoroughly. The caption mentioned, “such as repairing this statue, which can easily obtain the power of fortune, and it can be realized by wearing earrings, scriptures, victory flags, and lotus flowers for later generations”. This is a simple caption of the image and a profound interpretation of the information in the Thangka image, making the Thangka caption content more abundant.

In Figure 2b, the Thangka caption generated by Clipcap was insufficient, only describing a few attribute features of the Thangka image. In contrast, the caption generated by the proposed method was almost identical to the manually annotated caption, indicating that the proposed method has high accuracy. Additionally, the proposed method avoided obvious grammatical errors, and the generated caption was accurate regarding grammar and semantics. In addition, although the caption of the Thangka generated by the method proposed in this paper is richer in content, there are also cases of misdescription due to the complexity of the images themselves, as shown in Figure 2c.

### 4.5. Ablation Experiments

In this section, we conduct ablation experiments to investigate how each SCAMF-Net component affects the Thangka dataset’s overall performance. Design the following ablation experiments using the idea of controlling variables:

As shown in Table 3, the results started from the traditional Transformer-based encoder–decoder structure (baseline), which only used CLIP as the encoder–decoder for images and text for model training. With the introduction of the SCP and MFO modules (the Filter and Captioner), the model improved on all metrics, where a higher score indicates that the generated Thangka image caption is closer to the manually annotated caption in the dataset, and the generated caption effect is better. Specifically, after introducing the SCP method, we set the context length to 4, 8, and 16. The experimental results showed that the shorter context vector space in the SCP method had a higher BLEU-4 indicator, which was beneficial for the generalization of the image captioning model. The more extended context vector space had a higher CIDEr evaluation indicator, which could obtain more semantic information to make the generated caption more consistent with the reference. In addition, after adding the Captioner, the BLEU-4, METEOR, ROUGE, CIDEr, and SPICE metrics were improved by 5.6%, 5.8%, 6.3%, 55.3%, and 4.7%, respectively. This was because the Captioner optimized the data source with semantically rich synthetic captions, thereby guiding the model to generate higher-quality Thangka captions. After adding the Filter, the metrics were improved by 4.8%, 4.7%, 5.4%, 50.4%, and 3.7%, respectively. This was because the Filter removed noisy data during the training process and guided the model to train in the correct direction. Finally, by combining all modules into the baseline model, the best performance was achieved, with BLEU-4, METEOR, ROUGE, CIDEr, and SPICE improved by 8.7%, 7.9%, 8.2%, 76.6%, and 5.7%, respectively, further demonstrating the positive effectiveness of the method proposed in this paper.

## 5. Conclusions

To address the issues of low accuracy and insufficient content richness in deep learning-based image captioning methods for Chinese captions of Thangka images, this paper proposes SCAMF-Net, based on the contrastive language–image approach. First, the SCP module of SCAMF-Net concatenates context prompts with text features, enabling the image captioning model to learn more Thangka information in visual concept learning, thereby improving the richness of the generated text. Second, we designed the MFO module to further optimize the embedding space of the contrastive language image and improved the correlation between images and texts through the Captioner and Filter, thereby making the model generate more accurate captions. Compared with existing methods, our approach achieved good results on various evaluation metrics on the Thangka and COCO datasets, generating more reasonable and rich description content. Both objective and subjective evaluations confirmed that our method has excellent performance.

## Figures and Tables

**Figure 1 jimaging-09-00162-f001:**
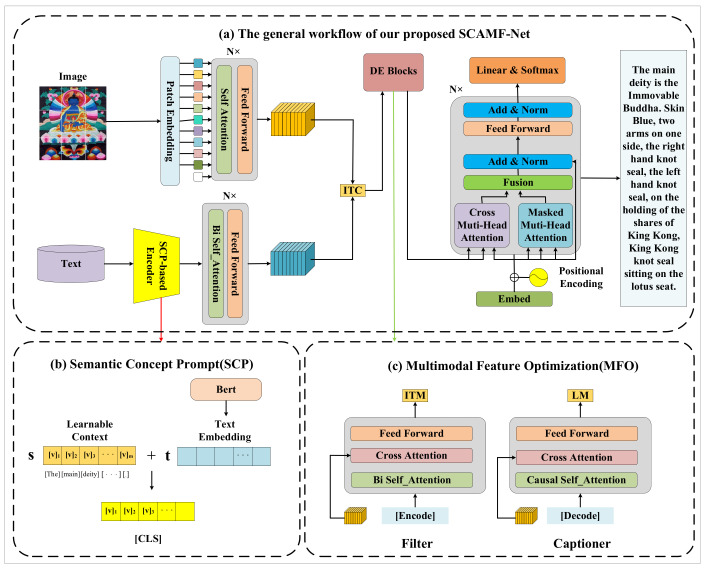
The proposed SCAMF-Net can efficiently convert Thangka images into Chinese captions: (1) Semantic Concept Prompt (SCP) is adopted to model context words of prompts using learnable vectors, and (2) Multimodal Feature Optimization(MFO) is introduced to remove noise data by Filter and Captioner.

**Figure 2 jimaging-09-00162-f002:**
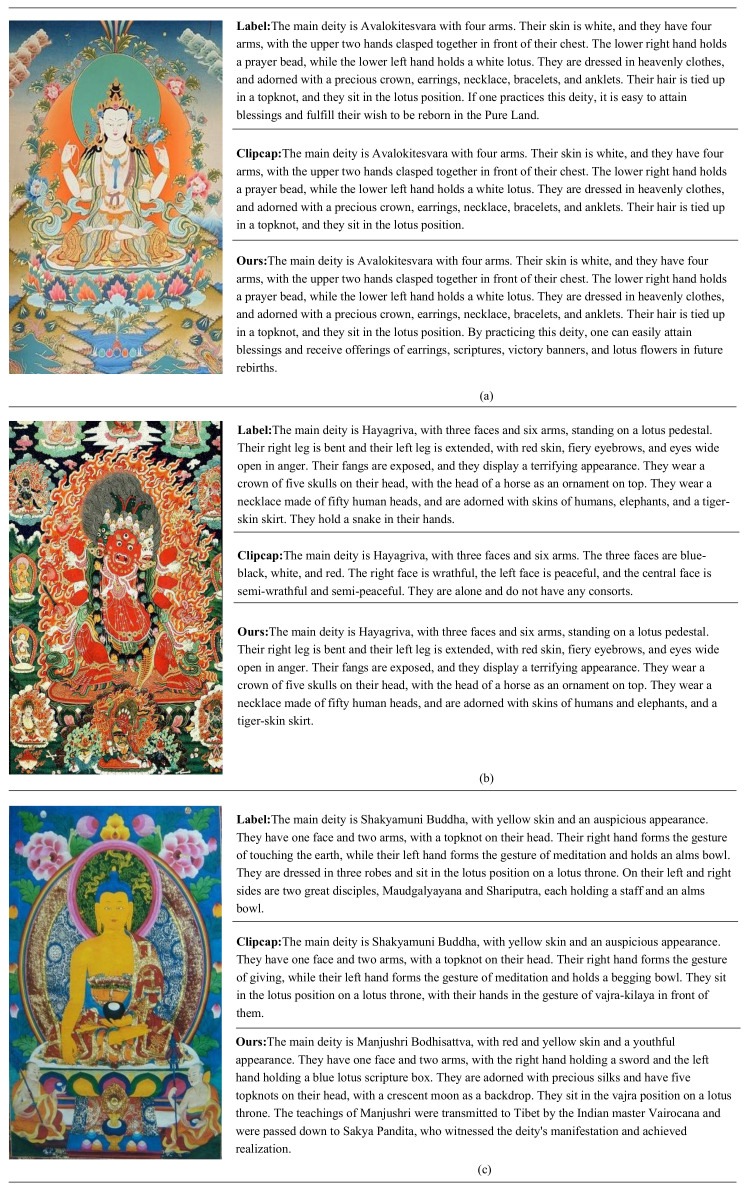
Comparison of experimental results between the mainstream method and the proposed method. Our SCAMF-Net achieves state-of-the-art image caption performance by generating more accurate and richer captions from Thangka mural images (Map of the caption in English).

**Table 1 jimaging-09-00162-t001:** The quantitative evaluation results show that the proposed SCAMF-Net is superior to Bi-LSTM, BUTD, VLP, and ClipCap on the Thangka dataset (unit %).

Model	BLEU-1	BLEU-4	CIDEr	METEOR
Bi-LSTM [45]	42.9	36.3	299.7	28.7
BUTD [13]	48.8	24.3	268.5	33.9
VLP [20]	58.8	44.8	444.6	44.5
ClipCap [22]	51.1	48.4	390.6	38.0
SCAMF (VGG)	45.5	28.3	286.4	29.4
SCAMF (Inception)	46.5	30.3	318.6	30.6
SCAMF (EfficientNetV2)	47.2	36.4	341.2	34.5
SCAMF (Realformer)	60.5	58.6	484.5	46.3
SCAMF (T5)	64.5	62.3	540.6	49.4
SCAMF-Net	70.5	63.6	562.4	52.0

**Table 2 jimaging-09-00162-t002:** The quantitative evaluation results show that the proposed SCAMF-Net is superior to BUTD, MMbert, ViLBert, LXMERT, UNITER, AoA, VLP, CLIP, Oscar, etc. on the COCO dataset (unit %).

Model	BLEU-4	METEOR	CIDEr	SPICE
BUTD [13]	36.3	27.7	120.1	21.4
MMBert [16]	39.5	29.4	130.6	22.6
ViLBert [17]	37.3	27.9	122.3	20.9
LXMERT [18]	37.8	26.7	124.8	21.4
UNITER [19]	38.4	28.7	128.4	21.8
VLP [20]	39.5	29.3	129.8	22.4
CLIP [21]	38.6	28.8	127.9	22.7
ClipCap [22]	33.5	27.4	113.1	21.1
AoA [43]	38.9	29.1	119.8	20.4
Oscar [44]	36.5	30.3	123.7	23.1
LEMON [46]	40.3	30.2	133.3	23.3
SCAMF-Net	40.7	31.3	133.5	23.9

**Table 3 jimaging-09-00162-t003:** The results of the ablation experiment on the Thangka dataset validated the effectiveness of the proposed methods (unit %).

Model	BLEU-4	METEOR	ROUGE	CIDEr	SPICE
Baseline	54.9	44.1	71.3	485.8	41.8
Baseline + SCP (M = 4)	57.5	43.7	61.7	443.5	40.3
Baseline + SCP (M = 8)	61.7	50.6	67.2	540.1	43.2
Baseline + SCP (M = 16)	60.7	49.6	62.2	550.1	45.2
Baseline + MFO (Captioner)	60.5	49.9	77,6	541.1	46.5
Baseline + MFO (Filter)	59.7	48.8	76.7	536.2	45.5
SCAMF-Net	63.6	52.0	79.5	562.4	47.5

## Data Availability

The datasets used in this study are available upon request from the email: wenjin_zhm@126.com.

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
