# Peer review of "Thangka Image Captioning Based on Semantic Concept Prompt and Multimodal Feature Optimization"

_2313-433X, 2023, doi:10.3390/jimaging9080162_

Round 1
Reviewer 1 Report
Thangka Image Captioning Based on Semantic Concept Prompt and Multimodal Feature Optimization
This paper presents a multi-layer cross-genre corpus, containing causal relations, temporal to leverage broader textual characteristics and semantic information.
Have you tried different backbones rather than VIT? For example efficientnet v2 or similar? The detectors need broader literate review, many missing, resnet backbone, etc. There are many papers that address this problem that are not listed, as example MMbert, and others.
Realformer can be successfully used in combination with backbone and contrastive learning. There are others, such as VilBert, or Uniter.
What about T5 with a backbone? It can lend interesting results.
Have you tried contrastive supervised loss with jacquard similarity? Its know to improve captions in comparison with SimCLR? How the proposed loss compares to this?
Label soothing? Is employed?
the Semantic Concept Prompt (SCP) can be seen as a downstream task?
The small size of Thank dataset, in terms of image or text pairs? Or both?
Line 276 - contrastive is bad written
More examples of the results(in English are needed)
Is it possible to have a map of the caption in English, Im, not a Chinese guy, but It would be nice. Perhaps as a second task, generate the caption in a different language ????
More mathematical background on the multimodal transformer is needed.
The English is ok, More reduced to the point sentences are welcome.
Some typos are easily found using online tools such Grammarly
Author Response
Comments from reviewer 1:
Comment#1: This paper presents a multi-layer cross-genre corpus, containing causal relations, temporal to leverage broader textual characteristics and semantic information.
Response#1: Thank you for taking the time to review our work. We are happy to hear your feedback on our paper and as well as the recognition of our work.
Comment#2:Have you tried different backbones rather than VIT? For example efficientnet v2 or similar? The detectors need broader literate review, many missing, resnet backbone, etc. There are many papers that address this problem that are not listed, as example MMbert, and others.
Response#2: We sincerely thank you for this detailed comment. Yes, we understand the importance of testing different architectures as model backbones, and we have tried several such as VGG, Inception, MobileNet, ResNet, and EfficientNet V2. However, these traditional methods do not perform well in describing the content of Thangka images. Our related work has also listed some multimodal methods similar to MMbert, but they cannot solve the problems in Thangka description very well. This is mainly due to the characteristics of the Thangka dataset itself: the images cover a wide range of subject matter and contain rich cultural connotations and domain-specific knowledge, which are abstract information that is difficult to capture.
Manuscript changes – We added experimental comparisons with several traditional backbone networks in Table 2 and further described our rationale for choosing this benchmark model by including a description of multimodal methods such as MMbert, ViLBert, LXMERT, and UNITER in (lines 121-132).
Comment#3:Realformer can be successfully used in combination with backbone and contrastive learning. There are others, such as VilBert, or Uniter.
Response#3: We sincerely thank you for raising this question. This is very helpful for our research, as we have improved the text processing part of our model by using Realformer's self-attention mechanism.
Manuscript changes – In Table 2, we added comparative experiments for MMBrty, VilBert, LXMERT, and Uniter, and in Table 1 further conducted comparative experiments by introducing Realformer.
Comment#4: Realformer can be successfully used in combination with backbone and contrastive learning. There are others, such as VilBert, or Uniter.
Response#4: We sincerely thank you for raising this question.
Manuscript changes – In Table 1, we added experiments using T5 as the backbone.
Comment#5:Have you tried contrastive supervised loss with jacquard similarity? Its know to improve captions in comparison with SimCLR? How the proposed loss compares to this?
Response#5: We sincerely thank you for raising this question. Yes, we have tried using Jaccard similarity for the contrastive loss, but the descriptive performance was not satisfactory. Compared to SimCLR, Jaccard similarity has some limitations in multi-modal image-text contrast. First, Jaccard similarity does not consider the importance or weight of elements, which means it cannot distinguish the importance of different features in images or words in text. Second, Jaccard similarity does not consider the order information between elements, which may ignore the structure and semantic relationships between images and text. The proposed loss function can utilize deep learning models to learn more representative feature representations, handle more complex multi-modal data, and achieve better experimental results.
Comment#6:Label soothing? Is employed?
Response#6: We sincerely thank you for raising this question. Yes, we used label smoothing to reduce the model's excessive confidence in positive and negative samples during the training process. We used label smoothing of 0.1 when calculating the loss.
Comment#7:the Semantic Concept Prompt (SCP) can be seen as a downstream task?
Response#7: We sincerely thank you for raising this question. Yes, Semantic Concept Prompt(SCP) can be seen as a downstream task that fine-tunes the pre-trained model to adapt to specific features and requirements of the Thangka image caption.
Comment#8:The small size of Thank dataset, in terms of image or text pairs? Or both?
Response#8: We sincerely thank you for raising this question. The Thangka dataset has relatively few image-text pairs, and even after data augmentation, the dataset only contains 3974 images with a corresponding single description for each image.
Comment#9:Line 276 - contrastive is bad written
Response#9: We sincerely thank you for raising this question.
Manuscript changes – We added a detailed description of the Image-Text Contrastive Loss on page 7 (lines 305-308) to make our paper more reader-friendly.
Comment#10:More examples of the results(in English are needed)
Response#10: We sincerely thank you for raising this question.
Manuscript changes – We added more examples of the results in Figure 2 ,and added a map of the caption in English in figure 3.
Comment#11:Is it possible to have a map of the caption in English, Im, not a Chinese guy, but It would be nice. Perhaps as a second task, generate the caption in a different language ????
Response#11: We sincerely thank you for raising this question. As Thangka images embody rich traditional Chinese culture, such as various Buddhas, bodhisattvas, arhats, and special textual symbols that carry their own symbolic meanings, they have their own cultural characteristics. captions in other languages may not accurately express their content, so generating captions in different languages is not currently considered as a second task.
Manuscript changes – we added a map of the caption in English in figure 3.
Comment#12:More mathematical background on the multimodal transformer is needed.
Response#12: We sincerely thank you for raising this question.
Manuscript changes – We added the mathematical background of the multimodal transformer. (lines 253-262)
Comment#13:The English is ok, More reduced to the point sentences are welcome.
Response#13: We sincerely thank you for raising this question.
Manuscript changes – We corrected some grammar errors and streamlined the content to make the article more concise.

Reviewer 2 Report
All-in-all a very novel and interesting project. The content of the paper is good: it is well structured, with sound scientific reasoning and methodologies. Culturally, I think it is of great merit. I know little about the specific language and cultural context however, but I think the NLP and image-based application is very interesting.
I have one critique of the paper, which is the Results. My understanding is that the authors compared to one baseline methodology and added components of the encoder-decoder methodology, eventually building into their own. This is a very interesting way to show the robustness of a new approach, so credit is due there.
However, is the baseline the only one methodology for comparison? I would want to see more comparisons to others. It may be the case that there are few, or that the generic component method is what is being compared to, which is fine. But more evaluation should be given there.
I also note that the accuracy of the methods is globally very low. Even if the author's methods are improving what currently is there, I think more analysis of that is needed. It may be the case that accuracy for these datasets is generally low. But it is always a point of contention, when it is universally the case, in my opinion. Finally, I did not understand the units being employed in the CIDEr comparison, so some explanation of that would be good too.
Author Response
Comment#1:All-in-all a very novel and interesting project. The content of the paper is good: it is well structured, with sound scientific reasoning and methodologies. Culturally, I think it is of great merit. I know little about the specific language and cultural context however, but I think the NLP and image-based application is very interesting.
Response#1: Thank you for taking the time to review our work. We are delighted to hear that you found our paper to be well structured, with sound scientific reasoning and methodologies. Your positive feedback is greatly appreciated.
Comment#2:I have one critique of the paper, which is the Results. My understanding is that the authors compared to one baseline methodology and added components of the encoder-decoder methodology, eventually building into their own. This is a very interesting way to show the robustness of a new approach, so credit is due there.
Comment#3:However, is the baseline the only one methodology for comparison? I would want to see more comparisons to others. It may be the case that there are few, or that the generic component method is what is being compared to, which is fine. But more evaluation should be given there.
Response#2&3: We sincerely thank you for raising this question. We added a comparison of our proposed method with other methods on the Thangka dataset in Table 1, and in Table 2, we also added a comparison with some state-of-the-art methods on the MSCOCO public dataset to further evaluate the effectiveness of our method.
Manuscript changes – In Table 2, we added comparative experiments for MMBrty, VilBert, LXMERT, and Uniter.
Comment#4:I also note that the accuracy of the methods is globally very low. Even if the author's methods are improving what currently is there, I think more analysis of that is needed. It may be the case that accuracy for these datasets is generally low. But it is always a point of contention, when it is universally the case, in my opinion. Finally, I did not understand the units being employed in the CIDEr comparison, so some explanation of that would be good too.
Response#2: We sincerely thank you for raising this question.We added an analysis of the generally low accuracy of image captioning and an explanation of CIDEr.
Manuscript changes – We added an explanation for the generally low accuracy of image captioning on the first page (lines 27-31), and provided an explanation for the unit of CIDEr on page 9 (lines 380-384). In Tables 1, 2, and 3, we added the unit of measurement (unit %) for the evaluation metrics.

Reviewer 3 Report
This paper seems interesting to me since it proposed a new image captioning model using concept prompts. The idea is interesting. I only have several minor concerns about this paper.
1) For the experiments on COCO dataset, the authors need to add more recent advanced works for comparison.
2) It is better to visualize or show what the learnable context token [v] learns.
3) I found this paper shares the similar idea with a recent work Switchable Novel Object Captioner, TPAMI 2023, where both works use the learnable prompts to indicate a possible object for the image captioning. The authors should discuss the difference from the existing work.
4) It is better to show some failure cases in fig. 2.
needs to improve
Author Response
Comment#1: For the experiments on COCO dataset, the authors need to add more recent advanced works for comparison.
Response#1: We sincerely thank you for raising this question.
Manuscript changes – In Table 2, we added comparative experiments for MMBrty, VilBert, LXMERT, and Uniter.
Comment#2:It is better to visualize or show what the learnable context token [v] learns.
Response#2: We sincerely thank you for raising this question.
Manuscript changes – We added an explanation of the learned context token [v] in Figure 1.
Comment#3: I found this paper shares the similar idea with a recent work Switchable Novel Object Captioner, TPAMI 2023, where both works use the learnable prompts to indicate a possible object for the image captioning. The authors should discuss the difference from the existing work.
Response#3: We sincerely thank you for raising this question.
Manuscript changes – We discussed the differences between the Switchable Novel Object Captioner and existing works on page 4 (lines 167-172).
Comment#4: It is better to show some failure cases in fig. 2.
Response#4: We sincerely thank you for raising this question.
Manuscript changes – We added failure cases in Figure 2.

Round 2
Reviewer 1 Report
I would suggest to include the results of those that are not soo good backbones, and list why the results are not those expected.
Minor typos
Author Response
Comment#1:I would suggest to include the results of those that are not soo good backbones, and list why the results are not those expected.
Response#1: We sincerely thank you for this detailed comment.
Manuscript changes –We added comparative experiments with VGG, Inception, and EfficientNet V2 in Table 1, and analyzed the impact of different architectures as the backbone of the model on the experimental results on pages 10 (lines 398-407) and corrected some typos in the article.

Reviewer 3 Report
Improve the English if possible
The revised version well addressed my concerns.
Author Response
Comment#1:Improve the English if possible
Response#1: We sincerely thank you for raising this question.
Manuscript changes – We have corrected some grammar errors and made revisions to enhance the content and presentation of the article.
